# OpenReview forum: "Deceive, Detect, and Disclose: Large Language Models Playing Mini-Mafia"
_ICLR.cc/2026/Conference — Submitted to ICLR 2026_

### Official Review · Reviewer_51Pd · 2025-10-26

**Soundness:** 3
**Presentation:** 3
**Contribution:** 2
**Rating:** 4
**Confidence:** 4

**Summary:**

This paper introduces Mini-Mafia, a simplified four-player version of the social deduction game Mafia, as a benchmark for evaluating LLMs on three interactive capabilities: deception, deception detection, and strategic information disclosure. The authors design controlled multi-agent experiments where LLMs play fixed roles (mafioso, detective, villager) in structured discussions, isolating each capability via role-specific win conditions. A standardized two-stage evaluation procedure estimates win rates and aggregates model performance across backgrounds. Results show smaller models sometimes outperform larger ones, and reveal social phenomena such as name bias and last-speaker advantage. The authors suggest the benchmark also aids AI safety research by generating data for deception detection and tracking deceptive tendencies in LLMs.

**Strengths:**

1. Well-structured experimental framework isolating interactive capabilities under controlled conditions.
2. Additional analyses (e.g., name bias, last-speaker advantage) provide meaningful insights into emergent social dynamics.

**Weaknesses:**

1. Prior works have already explored deception games and benchmarks, such as Avalon and Werewolf. However, the paper does not clearly explain how Mini-Mafia differs from these existing setups or why this new variant is needed. The motivation for introducing a separate benchmark remains unconvincing.
2. Since the ultimate objective of such research is to enable LLMs to interact effectively with humans, the absence of any human-involved experiments significantly limits the benchmark’s practical relevance.
3. The setup may oversimplify real-world social reasoning, only one discussion round and fixed night actions limit depth of interaction.
4. Benchmark outcomes (win rates) depend heavily on prompt phrasing and turn order, which may confound interpretability.
5. No ablation or sensitivity study to confirm robustness against prompt wording or memory design.

**Questions:**

1. How does Mini-Mafia differ from existing deception-game benchmarks such as Avalon or Werewolf, and why is a new variant necessary?
2. If the ultimate goal is to enable LLMs to interact effectively with humans, why are there no human-involved experiments to validate the benchmark’s real-world relevance?
3. To what extent do prompt phrasing and turn order influence the benchmark’s outcomes?

---

> ### Author Response · Authors · 2025-11-24
> **Theoretical Framework and Robustness Studies**
>
> We thank the reviewer for their thoughtful and constructive feedback. Many of the concerns raised motivated substantial improvements to the paper. Below we describe the main additions and how they address the specific points raised.
>
> ## 1. Major Revision: Theoretical Breakthrough
> The reviewer's concerns about oversimplification and interpretability motivated a fundamental theoretical development. In Section 2.4 "Theoretical Framework," we introduce and validate a simple analytical model that accurately predicts mafia win probabilities:
> $$
> \text{logit}(p) = v \times (m - d),
> $$
> where $m$ is the mafioso deception capability, $d$ is the detective disclosure capability, and $v$ is the villager detection sensitivity.
>
> This transforms Mini-Mafia from merely another deception game into a benchmark with complete theoretical foundations. The model predicts all $I^3$ game outcomes from only $3I$ intrinsic parameters inferred via Bayesian inference, demonstrating that the simplification enables theoretical tractability rather than limiting it. We validate this model through the background-based analysis presented in the first version, now moved to Appendix B, finding highly consistent rankings.
>
> ## 2. Responses to Specific Questions
>
> ### Q1: How does Mini-Mafia differ from existing benchmarks and why is it needed?
>
> Key differentiator: Mini-Mafia is the first deception game benchmark with a validated theoretical model decomposing game outcomes into intrinsic role-specific capabilities. Prior work (Avalon, Werewolf) focuses on realistic gameplay complexity but lacks this theoretical foundation.
>
> Why this matters:
> - Interpretability: Parameters $(m,v,d)$ have precise mathematical definitions through their relationship to win probabilities (Eq. 1), not just relative rankings.
> - Predictive power: The model predicts outcomes across all model combinations from intrinsic parameters.
> - Scientific understanding: We can study how elementary capabilities measured here combine and scale in more complex social interactions.
>
> The simplification is not a limitation but a feature that enables theoretical understanding impossible in complex settings. Section 6.1 explicitly positions Mini-Mafia as measuring the elementary capabilities that serve as building blocks for understanding richer dynamics.
>
> ### Q2: Why no human-involved experiments?
>
> We agree this is valuable future work and explicitly acknowledge this limitation in Section 6.3 "AI Safety."  We clarify that the current work establishes the measurement methodology and theoretical foundations needed for future human comparison studies. We are working on to develop an online platform for collecting human gameplay data.
>
> ### Q3.1: Influence of prompt phrasing and turn order?
> In the added Robustness Appendix (new Appendix C) we tested extreme variations for ablation:
> - Minimal prompt: Reduced from detailed 800-word prompt to minimal 12-word prompt
> - Extended interaction: Increased from 2 to 8 discussion rounds
>
> Model rankings remain broadly consistent across these very different conditions, demonstrating that measurements capture fundamental capabilities rather than implementation artifacts.
>
> ### Q3.2: Influence of turn order?
> We explicitly quantify last-speaker advantage in Section 4.2 as an emergent phenomenon.
>
>
>
> ## 3. Addressing Weaknesses
>
> ### W1: Motivation for new variant
> As explained in Q1, the theoretical model (Eq. 1) provides the key differentiator. The simplification enables discovery of an analytical form decomposing complex multi-agent outcomes into interpretable, quantifiable capabilities.
>
> ### W2: Human involvement
> Addressed in Q2. We now explicitly discuss this as future work (Section 6.3) while clarifying that the theoretical framework enables meaningful human comparison when such data becomes available.
>
> ### W3: Oversimplification
> The theoretical model demonstrates this is a feature, not a bug. The parameters $(m,d,v)$ represent elementary forms of deception, detection, and disclosure that compound in more complex scenarios. Additionally, robustness studies (Appendix C) show results persist with 8 discussion rounds and short prompt.
>
> ### W4: Prompt and turn order dependence
> Directly addressed in Q3 with robustness studies. We also added strategic analysis in Section 3 explaining why models succeed or fail in each role, providing deeper interpretability beyond win rates.
>
> ### W5: No ablation studies
> Now included as Appendix C with extreme variations in prompt design and interaction length, demonstrating robustness of capability measurements.
>
> ## 4. Additional Improvements
>
> - Strategic analysis (Section 3): We added detailed analysis of winning strategies for each role:
> - Enhanced visualization: New Figure 1 shows gameplay flow and benchmark pipeline from games to parameter estimation.
> 5, and Figure 6 explains the old background-based methodology that we use for comparison and validation.

---

> > ### Comment · Reviewer_51Pd · 2025-11-25
> > **Thanks for the rebuttal**
> >
> > I have read the author rebuttal and the comments from other reviewers. My concerns remain, especially about the necessity of the simplified social-deception game and the motivation for formulating it as a simple theoretical model. I will maintain my ratings.

---

> > > ### Author Response · Authors · 2025-11-26
> > > **Clarification on Theoretical Contribution**
> > >
> > > Thank you for your consideration.
> > >
> > > We want to clarify that the theoretical model $\text{logit}(p)=v\times (m−d)$ is not just a reformulation but a significant empirical discovery. To the best of our knowledge, this is the first example of a three-agent system where probabilistic outcomes are described by an analytical function (the Fermi-Dirac distribution from quantum physics!) depending on intrinsic agent parameters. This predictive framework, extensible to richer social-deduction variants and absent in prior benchmarks,  justifies both the necessity and motivation for our simplified variant.
> > >
> > > We respect that this may not fully address your concerns.

---

> ### Author Response · Authors · 2025-12-02
> **Human baseline**
>
> Complementing our new theoretical advances and directly addressing weakness 2, we added human data to the benchmark. These results show that while LLMs can excel at persuasive communication, they still lag behind humans in the strategic reasoning required for agentic interaction.
>
> We believe these addition significantly strengthen the paper's contribution.

---

### Official Review · Reviewer_6FWk · 2025-10-31

**Soundness:** 3
**Presentation:** 3
**Contribution:** 2
**Rating:** 2
**Confidence:** 3

**Summary:**

The paper proposes Mini-Mafia, a simplified version of the Mafia game, to evaluate the ability of LLMs to deceive, detect deception, and disclose information. These three skills directly map to the characters in the game. At a single instance, models are assigned a role in the game, play against each other and the win rates are the aggregated across multiple game configurations. The paper also reports interesting phenomena such as name bias and last-speaker advantage.

**Strengths:**

The mini-mafia benchmark provides a structures and reproducible way of measuring the three skills deceive, detect, and disclose with a clear mapping to the roles of the game. A large number of 100 games are run for each configuration and the analysis via hierarchical bayesian approaches is rigorous.

The emergent results of name bias and last-speaker advantage are interesting observations. The full prompts used to run the game are provided which helps reproducibility. The paper is overall well written and clear.

**Weaknesses:**

My primary concerns are the limited scale of the environment, applicability of the benchmark to the same skills in real-world scenarios and scope of the evaluation.

**Limited scale of the environment:**

The environment is limited to only 4 players and a single day phase (the night phase is fixed). This simplifies the social dynamics of the mafia game a lot. Extending the benchmark to multiple day-night phases, with a non fixed night phase and also scaling the number of players would significantly strengthen the paper. These extensions would allow for longer term strategies and more interesting dynamics.

**Real-world applicability of the benchmark:**

I am doubtful whether the insights obtained regarding the skills deceive, detect, and disclose can be transferred to any real-world scenarios. The current version of the game seems to oversimplify the complex dynamics and role descriptions that may be encountered in real-world tasks. This also raises another question whether win rates in the mafia game can be interpreted as the model's ability to deceive / detect / disclose?

**Evaluation scope:**

A size scaling study within multiple families would be helpful to further understand the correlation between size and the evaluated skills. Experiments scaling the number of discussion steps would be very interesting (and as mentioned before number of agents and day/night phases).

**Other points:**

* I liked section 4 with broader trends that were observed. However, section 3 mostly describes the tables and figures on a high level rather than providing a deeper analysis.
* Is the name bias really significant? The differences seem to be small.
* While the last-speaker advantage is an interesting observation, I find it too strong to conclude that procedural elements significantly impact social outcomes.
* How are the different backgrounds affecting the results? Across three different skills three different backgrounds were shown which makes it harder to compare.
* Typo: line 325 "Compare..."

Overall, I think this work is promising, but needs some more work (especially regarding the scale of the environment).

**Questions:**

I would be happy to increase my score if the authors can address some of the following points:
* How would results change if multiple day–night cycles were allowed, or if the detective could choose not to disclose immediately?
* How would results be affected by a longer discussion phase during the day?
* Why was the number of agents limited to 4? Have the authors tried scaling up?
* Could the benchmark be extended to cooperative dynamics (e.g., villagers coordinating or sharing partial information)?
* Could the authors show correlations between model size, reasoning style, and success rates?
* Regarding the last-speaker advantage: can it be said that the more information the model has the larger the last-speaker advantage?

---

> ### Author Response · Authors · 2025-11-24
> **Theoretical Framework and Robustness Studies**
>
> We thank the reviewer for their thoughtful and constructive feedback. Many of the concerns raised motivated substantial improvements to the paper.
>
> ## 1. Major Revision: Theoretical Breakthrough
>
> The reviewer's concerns about limited scale, real-world applicability, and whether win rates meaningfully measure capabilities motivated a fundamental theoretical development. In Section 2.4 "Theoretical Framework," we introduce and validate a simple analytical model:
> $$
> \text{logit}(p) = v \times (m - d),
> $$
> where $m$ is the mafioso deception capability, $d$ is the detective disclosure capability, and $v$ is the villager detection sensitivity.
>
> This directly addresses the core concern about win rate interpretability: the parameters $(m,d,v)$ are mathematically defined quantities with precise relationships to observable win probabilities. The model predicts all $I^3$ game outcomes from only $3I$ intrinsic parameters inferred via Bayesian inference, demonstrating that win rates can indeed be decomposed into interpretable capability measurements.
>
> We validate this model through the background-based analysis from the first version, now moved to Appendix B, finding highly consistent rankings between both methodologies.
>
> ## 2. Addressing Specific Weaknesses
>
> ### W1: Limited Scale of Environment
>
> - **Robustness studies** (new Appendix C): We tested extended interactions with 8 discussion rounds. Model rankings remain broadly consistent.
> - **Theoretical justification:** The discovery of the analytical model (Eq. 1) demonstrates that the simplification enables theoretical tractability.
> - **Design philosophy**: Mini-Mafia is designed to isolate and measure fundamental capabilities. Just as physics studies simple systems before complex ones, we measure elementary social capabilities before tackling their composition in multi-round, multi-player settings.
>
> ### W2: Real-World Applicability
>
> - **Generality through decomposition**: The parameters $(m,d,v)$ capture elementary interaction mechanisms that appear across contexts, not just Mafia-specific strategies. For example: $m$ measures quality of counter-narrative execution when caught in deception, $v$ measures sensitivity to gaps between competing claims, and $d$​ measures effectiveness of information disclosure under adversarial pressure.
> - **Empirical evidence of generality**: The robustness studies (Appendix C) show these capabilities persist across extreme variations in prompt design (800-word detailed vs. 12-word minimal) and interaction length (2 vs. 8 rounds).
> - **Future work pathway (Section 6.1)**: We explicitly discuss extending to richer scenarios as studying how elementary capabilities measured here combine in more complex real-world interactions.
>
> ### W3: Evaluation Scope
>
> - **Size scaling within families**: We agree that would be nice to add family of models of different sizes. We plan to include Claude Haiku for comparison. While we don't have complete size-scaling studies, our results already show striking patterns, as Claude Sonnet 4 performing the worst in detection.
>
> - **Discussion rounds scaling**: Now directly addressed in Appendix C with 8-round experiments.
>
> ### W4: Other points
>
> - **Section 3 Analysis Depth**: We substantially strengthened Section 3 with strategic analysis explaining why models succeed or fail:
>
> - **Name Bias Significance**: The name bias differences are statistically significant, Bob is about $4\sigma$ above Diana. We now interpret this through the theoretical framework (Eq. 3).
>
> - **Last-Speaker Advantage Interpretation**: We softened the language but maintain this as a robust finding.
>
> - **How Backgrounds Affect Results**: The complete raw data and plots of win rates in every background is included in Appendix E.
>
> ## Responses to Specific Questions
>
> ### Q1: Multiple day-night cycles or delayed disclosure?
> Future work (Section 6.1) discusses multi-round variants.
>
> ### Q2: Longer discussion phases?
> Directly addressed in Appendix C with 8-round experiments showing consistent rankings.
>
> ### Q3: Why 4 agents? Have you tried scaling?
> Section 6.1 discusses scaling as future work, positioning current measurements as elementary capabilities that serve as building blocks for understanding larger-scale dynamics.
>
> ### Q4: Cooperative dynamics extensions?
> This is valuable future work. The theoretical framework provides a foundation, we could introduce parameters for coordination capabilities and study how they interact with $(m,d,v)$.
>
> ### Q5: Correlations between size and success?
> Our studies reveal that size does not uniformly predict social capabilities. This finding is a key contribution.
>
> ### Q6: Last-speaker advantage and information?
> Interesting hypothesis. The detective (complete information) shows larger advantage than mafioso (partial information), while villager (no information) shows essentially no advantage.
>
> ## 4. Additional Improvements
> Enhanced visualization: New Figure 1 and Figure 6 shows gameplay flow and benchmark pipeline.

---

> > ### Author Response · Authors · 2025-12-02
> > **Human baseline**
> >
> > Complementing our new theoretical advances, we added human data to the benchmark. These results show that while LLMs can excel at persuasive communication, they still lag behind humans in the strategic reasoning required for agentic interaction.
> >
> > We believe these addition significantly strengthen the paper's contribution.

---

### Official Review · Reviewer_TgyV · 2025-11-03

**Soundness:** 2
**Presentation:** 2
**Contribution:** 2
**Rating:** 2
**Confidence:** 4

**Summary:**

The paper introduces an evaluation framework for assessing LLMs’ abilities to deceive, detect deception, and disclose truthful information using a minimal, single night setting of a Mafia game. The conversation are between one mafioso who has killed a villager, one detective who identifies the mafioso, and a villager, who decides which of the two to believe.

The authors evaluate 10 candidate models against 5 background models, with the candidate model as one role and background models as the other two. Each candidate–background pairing is rolled out 100 times, and win rates are estimated via Bayesian inference, yielding corresponding means and variances. Model performance is then aggregated by computing z-scores across backgrounds and candidates.

The results show that no single model dominates all dimensions and small models sometimes outperform bigger models. Additionally, the authors discovered biases in  name and gender bias for trust attribution, as well as a more pronounced advantage for the last speaker

**Strengths:**

I like the paper’s novel approach of reducing the Mafia game simulation to a minimal, single-night conversation. Most existing social-deduction-based benchmarks focus on realism and implement multiple turns with more partners and roles, and models performances are estimated through aggregate outcomes over games. While such setups capture richer dynamics, they often reduce interpretability and make it difficult to isolate model abilities from their partners. However, this minimal Mafia setting offers a more interpretable and controlled environment, enabling a cleaner and simpler analysis of each model’s capabilities.

I also like the evaluation methodology, which is enabled by this simplified setup. By aggregating z-scores across background models, the authors provide a simple and unbiased method to estimate a model’s ability in a given role. This approach is practical and accessible for the community to reproduce and extend similar analyses.

The paper is also fluent and well structured, and ideas are well communicated.

**Weaknesses:**

First, while the mini-Mafia setup is conceptually sound, it is overly simplified and provides limited coverage of realistic game dynamics. As a result, it offers shallow insight into the deception, detection, and disclosure abilities the paper aims to evaluate. Because the setting only covers the first night, each role has minimal context to leverage for either deception or detection. This leads to short, repetitive, and low-content interactions, as reflected in several examples shown by the authors. A key challenge of deception lies in maintaining consistency with and taking advantage of prior facts and statements, which this setup does not capture. Similarly, deception detection and truthful disclosure depend heavily on contextual and temporal reasoning, which are largely absent here. To make the evaluation informative of modern LLM enabled dialogue system, the framework should at least include additional background scenarios or multi-turn extensions to enable richer interaction histories.

Second, the paper lacks semantic analysis explaining why certain models perform better at deception, detection, or disclosure. A more insightful analysis could involve categorizing persuasion or deception strategies (e.g., exploiting communication gaps as noted in Section 3.1) and examining their distribution across models. It would also strengthen the work to analyze game dynamics, such as how strategies evolve or how the villager’s trust shifts over discussion.

Third, the presentation could use more illustrative graphics and visualizations, which would make the framework and findings easier to understand and provide richer analysis.

**Questions:**

1. Beyond interpretability, what do you see as the main advantages of your framework compared to more comprehensive, realistic social-deduction benchmarks?
2. Given the minimal context available to each role, what factors do you think make a model appear more persuasive in this setting? From the examples, most persuasion strategies seem limited to direct self-identification, while villagers’ reasoning appears biased by game constraints (e.g., questioning why the detective investigated the mafioso—a system-determined action). How do you plan to analyze or mitigate these biases and encourage more diverse persuasion strategies?
3. As I mentioned in weaknesses, I suggest the framework to include additional background scenarios or multi-turn extensions to enable richer interaction histories. The authors could sample more conversation trajectories, find interesting middle points that require more complex deception/persuasion/detection abilities as additional starting points of the discussions.

---

> ### Author Response · Authors · 2025-11-24
> **Simplicity as a Virtue: A Theoretical Breakthrough**
>
> We thank the reviewer for their thoughtful and constructive feedback. Many of the concerns raised motivated substantial improvements to the paper. Below we describe the main additions and how they address the specific points raised.
>
> ## 1. Major Revision: Theoretical Breakthrough
>
> The reviewer's concerns about limited insight and shallow coverage motivated a fundamental theoretical development that transforms the contribution of this work. In Section 2.4 "Theoretical Framework," we introduce and validate a simple analytical model that accurately predicts mafia win probabilities across all model combinations:
> $$
> \text{logit}(p) = v \times (m - d),
> $$
> where $m$ is the mafioso deception capability, $d$ is the detective disclosure capability, and $v$ is the villager detection sensitivity.
>
> This theoretical breakthrough substantially strengthens our contribution:
> - From descriptive to predictive. The model predicts all $I^3$ game outcomes from only $3I$ intrinsic parameters inferred from gameplay using Bayesian inference.
> - Simplicity enables theoretical tractability. The discovery of a compact analytical form demonstrates that our simplification enabled theoretical understanding rather than limiting it.
> - Independent validation. We validate the theoretical model through the background-based methodology from the original submission (now Appendix B), finding highly consistent rankings.
>
> ## 2. Addressing Specific Weaknesses
>
> ### 2.1 On Limited Coverage and Shallow Interactions
>
> - Empirical robustness studies (Appendix C): We tested extreme variations including minimal 12-word prompts (versus our detailed 800-word prompts) and 8 discussion rounds (versus 2 rounds). Model rankings remain broadly consistent, demonstrating that our measurements capture fundamental capabilities rather than implementation artifacts.
> - Theoretical justification: The theoretical model reveals that even this simplified setting isolates fundamental interaction mechanisms. The parameters $(m,d,v)$ represent elementary forms of deception, detection, and disclosure, building blocks that compound in more complex scenarios.
>
> ### 2.2 On Semantic Analysis of Strategies
>
> - We substantially strengthened Section 3 with strategic analysis.
>
> ### 2.2 On Absolute Baselines and Interpretability
>
> The reviewer was concerned that scores were relative and lacked intrinsic meaning. The new theoretical framework resolves this because parameters $(m, d, v)$ are directly connected to observable win probabilities through Eq. 1 and 2.
>
> ### 2.3 On Visualization
>
> - We added Figure 1: a two-panel figure showing (a) Mini-Mafia gameplay flow with role indicators and (b) benchmark pipeline from games to parameter estimation. We also reorganized appendices with additional comparison figures.
> - We added Figure 6 to Appendix B, illustrating the background-based methodology that we retain for comparison and validation.
> - We added, a representative figure for the mafioso, villager and detective in the game transcripts.
>
> ## 3. Responses to Specific Reviewer Questions
>
> ### Q1: Main advantages beyond interpretability?
>
> - Theoretical grounding: The theoretical model provides principled understanding of how capabilities combine, a pre-requisite to tackle complex, realistic settings.
> - Computational efficiency: Testing 10 models required only ~14,000 games versus orders of magnitude more for realistic benchmarks.
> - Decomposition of social intelligence: By isolating fundamental capabilities, we can study how they scale and combine in more complex scenarios.
>
> ### Q2: What makes models persuasive in this minimal setting?
>
> Our strategic analysis reveals that persuasiveness depends on execution quality rather than strategy diversity. The theoretical parameter $m_i$ captures the *quality* of deceptive execution—precisely the elementary capability we aim to measure as a building block for more complex interactions.
>
> ### Q3: On multi-turn extensions
>
> Robustness studies (Appendix C) show that extending to 8 rounds and shortening the prompt (to only 12 words!) maintains consistent rankings, suggesting our measurements capture capabilities that persist across interaction lengths. Section 6.1 discusses extending to richer Mafia variants as future work, positioning Mini-Mafia as establishing foundational measurements for understanding how capabilities combine in complex dynamics.
>
> ## 4. Additional Improvement
> Enhanced visualization: New Figure 1 shows gameplay flow and benchmark pipeline from games to parameter estimation. 5, and Figure 6 explains the old background-based methodology that we use for comparison and validation.

---

> > ### Author Response · Authors · 2025-12-02
> > **Human baseline**
> >
> > Complementing our new theoretical advances, we added human data to the benchmark. These results show that while LLMs can excel at persuasive communication, they still lag behind humans in the strategic reasoning required for agentic interaction.
> >
> > We believe these addition significantly strengthen the paper's contribution.

---

### Official Review · Reviewer_QmSc · 2025-11-11

**Soundness:** 2
**Presentation:** 2
**Contribution:** 2
**Rating:** 4
**Confidence:** 3

**Summary:**

This paper proposes a new benchmark based on the social deduction game Mafia. The benchmark is a simplified 4-player variant with fixed night actions that reduce the game to a single day-time discussion and voting phase. The benchmark is designed to measure three capabilities, the ability of an LLM to: deceive the other players, detect the hidden player (mafioso), and disclose the hidden player. These capabilities are evaluated across 10 models, with the other players fixed to another model (out of 5 possible "background" models). Results are shown which highlight some interesting results, e.g. last word advantage.

**Strengths:**

* I appreciate trying to distill a complex social reasoning game into a controlled, repeatable benchmark. I think this is a valuable goal, and while I'm not necessarily convinced that this is the solution to get there, I do think there is value in exploring this direction.

* The main empirical results are pretty thorough across target/background models. The full results in Appendix C show that a significant amount of effort went into running these matchups. I also like the identification of some of the emerging phenomena. While I'm not totally convinced of their rigor (see weakness below), any good benchmark paper benefits from highlighting things like this.

**Weaknesses:**

* The simplification of Mafia to a single day-time discussion eliminates much of the social complexity that was useful in the original game. Is there really much opportunity for deception or detection outside of the most simplistic behaviors ("No, Bob did it!"), when players don't have the ability to reason about prior events or behaviors? I am not convinced on the utility of what is actually being measured here, and think the game is *too* simple.

* These are all relative scores, and there's no way to measure an absolute baseline, e.g. human player performance.

* The "serendipitous results" (sec. 4) are not explored very thoroughly. In particular, the name bias is only present for one player (Diana) and the example given in the paper is not very convincing. That example seems like it would more likely be an example of last word advantage. Similarly, last word advantage may be related to sycophancy, but no further investigation is done here. All in all, I think there's a lack of interesting results/analysis here. What do we really learn from this benchmark? Why should other people in the community use it?

**Questions:**

1. In the example in Sec. 3.2, why would we assume this is name bias instead of last word advantage?

2. Is last word advantage associated with sycophancy? Do the other models just listen more closely to the last model which "spoke"?

---

> ### Author Response · Authors · 2025-11-23
> **Simplicity as a Virtue: A Theoretical Breakthrough**
>
> We thank the reviewer for their thoughtful and constructive feedback. Many of the concerns raised motivated substantial improvements to the paper. Below we describe the main additions and how they address the specific points raised.
>
> ## 1. Major Revision: Theoretical Breakthrough
>
> The reviewer expressed concerns about interpretability of our scores, lack of intrinsic meaning in relative rankings and whether the simplified game structure limits what we can learn. These points motivated a substantial theoretical development that is now central to the paper. In Section 2.4 "Theoretical Framework" we introduce and validate a simple analytical model that accurately predicts mafia win count $\omega\sim \text{Binomial}(n,p)$ from the theoretical model:
> $$
> \text{logit}(p) = v \times (m - d),
> $$
> where $m$ is the mafioso deception capability, $d$ is the detective disclosure capability, and $v$ is the villager detection sensitivity.
>
> This substantially strengthens our contribution in three ways:
> 1. **Intrinsic, role-specific parameters.** The parameters $(m,d,v)$ are no longer relative scores. They are mathematically defined by their relationship to the win rate.
> 2. **Predictive framework over all model triplets.**  The model predicts all $I^3$ game outcomes $p_{ijk}$ from only $3I$ parameters that can be inferred from gameplay.
> 3. **Theoretical tractability as a virtue of simplification.**  The discovery of a compact analytic form that fits the empirical outcomes shows that the simplification enabled theoretical understanding, rather than limiting it.
>
> We validate this theoretical model through the independent background-based analysis used in the first version, which is now in Appendix B, finding highly consistent rankings between both methodologies.
>
>
> ## 2. Addressing Specific Weaknesses
>
> ### W1 On Simplicity and Generalization
>
> The reviewer questioned whether Mini-Mafia is “too simple” and whether the results would generalize beyond the specific implementation.
> - **Robustness studies in Appendix C**, where we changed prompt and the number of discussion rounds. Using the background-based methodology that is now presented in Appendix B, we find that model rankings on the deceive dimension remain broadly consistent across these very different conditions.
> - **Design philosophy**: Mini-Mafia is designed to measure elementary forms of deception, detection, and disclosure. The simplicity is a feature, not a flaw: it enables analytical modeling and interpretable results. In future work, we aim to scale up to richer Mafia variants and study how these simple interaction capabilities compound and which new social dimensions emerge in more complex dynamics.
>
> ### W2 On Absolute Baselines and Interpretability
>
> The reviewer was concerned that scores were relative and lacked intrinsic meaning. The new theoretical framework resolves this because parameters $(m_i, d_j, v_k)$ are directly connected to observable win probabilities through Eq. 1 and 2.
>
> ### W3 On Serendipitous Results and Lack of Analysis
>
> The reviewer noted that the “serendipitous” results (name bias, last-speaker advantage) were interesting but under-analyzed. We strengthen this part substantially by interpreting the name bias in terms of new theoretical model parameters.
>
> ### W4 “What do we really learn, and why should others use this benchmark?”
>
> The reviewer asked what is fundamentally learned from Mini-Mafia and why others should adopt it. First, results are surprising: specially the fact that Claude Sonnet 4 is the worst detector, and that small models often perform better than large models. Besides that, the revision turns the paper from “just another benchmark” into a benchmark plus a theoretical characterization of the underlying multi-agent dynamics.
>
> ## 3. Responses to Specific Reviewer Questions
>
> ### Q1: Why assume name bias instead of last-word advantage?
>
> In Section 3.2, we use individual game transcripts only to *illustrate* phenomena, not as statistical evidence. The statistical claims come from aggregating across all $14.000$ games. But the example in the main text can illustrate both name bias and last-speaker advantage. The *disentangling* of these effects comes from the aggregated analysis.
>
> ### Q2: Is last-word advantage associated with sycophancy?
>
> We do not believe so. Sycophancy typically refers to models adapting their answers to agree with a user’s stated preferences or beliefs. This does not imply, per se, that one agrees with the last one to speak.
>
> ## 4. Additional Improvement
> - Enhanced visualization: New Figure 1 shows gameplay flow and benchmark pipeline from games to parameter estimation. 5, and Figure 6 explains the old background-based methodology that we use for comparison and validation.

---

> > ### Author Response · Authors · 2025-12-02
> > **Human performance**
> >
> > Complementing our new theoretical advances and directly addressing weakness 2, we added human performance to the benchmark. These results show that while LLMs can excel at persuasive communication, they still lag behind humans in the strategic reasoning required for agentic interaction.
> >
> > We believe these addition significantly strengthen the paper's contribution.

---

### Author Response · Authors · 2025-11-24
**Simplicity as a Virtue**

We have substantially revised the paper in response to reviewer feedback. Remarkably, we accomplished what was outlined in the original submission's Section 6.2 "Theoretical Extensions" as future work, a direction we initially believed would be extremely challenging. The key change is the introduction of a complete theoretical framework that transforms the contribution from descriptive benchmarking to predictive modeling of multi-agent social dynamics.

## Major Changes:

- **New Section 2.4 "Theoretical Framework"**: We introduce and validate an analytical model that predicts mafia win probabilities $p$ across all model combinations:
$$
\text{logit}(p) = v \times (m - d)
$$
This model demonstrates that complex multi-agent game outcomes can be decomposed into three intrinsic role-specific parameters: mafioso deception capability $(m)$, detective disclosure capability ($d$), and villager detection sensitivity $(v)$. The model predicts all $I^3$ game outcomes from only $3I$ parameters estimated via Bayesian inference.

- **Background-based analysis moved to Appendix B**: The original Section 2.4 "Statistical Analysis" (background-based methodology) has been relocated to Appendix B "Background-Based Analysis," where we use it as independent validation of the theoretical model. Both approaches produce highly consistent rankings, confirming the robustness of our measurements.

- **New Appendix C "Robustness Studies"**: We added ablation studies testing extreme variations (minimal 12-word prompts vs. detailed 800-word prompts; 2 vs. 8 discussion rounds) to demonstrate that our capability measurements are robust across implementation choices.

- **Enhanced strategic analysis (Section 3)**: We substantially strengthened the results section with detailed analysis explaining why models succeed or fail in each role, moving beyond descriptive statistics to mechanistic understanding.

- **Improved visualization**: We added new Figure 1 with two panels showing (a) Mini-Mafia gameplay flow with clear role indicators and (b) the complete benchmark pipeline from games to parameter estimation. We also added Figure 6 with a single panel explaining the complete background-based benchmark pipeline.

## Why This Matters:

The theoretical model addresses core reviewer concerns about interpretability, real-world applicability, and whether the simplified game provides meaningful insights. It demonstrates that the simplification enables theoretical tractability rather than limiting it, revealing elementary social capabilities that can be systematically measured and predicted. Following the scientific principle of starting with simple systems before tackling complex ones, Mini-Mafia measures these fundamental building blocks of social intelligence. Future work can extend to richer multi-player scenarios to study how these elementary capabilities compound and combine in more complex social dynamics.

We believe these revisions significantly strengthen the paper's contribution and directly address the concerns raised by all reviewers.

---

### Author Response · Authors · 2025-12-01
**Human Baseline**

Complementing our new theoretical advances and addressing a key reviewer critique, we added human data to the benchmark. These results show that while LLMs can excel at persuasive communication, they still lag behind humans in the strategic reasoning required for agentic interaction.

We believe these addition significantly strengthen the paper's contribution and directly address concerns raised by some reviewers.

---

### Author Response · Authors · 2025-12-02
**Summary for the new Area Chair**

With the hope to assist the new area chair we prepared this brief summary.

Across reviews, the main concerns were:
- Whether the **simplified Mini-Mafia** setting is sufficiently justified/meaningful;
- The absence of key **ablation studies**;
- The lack of a **human baseline**.

To address these, we made several substantive additions in our rebuttal materials:
- We validated a new **theoretical model**  that directly shows that simplicity is a desirable feature enabling **interpretability**. To the best of our knowledge, this is the first example of a three-agent system where probabilistic outcomes are described by an analytical function (the Fermi-Dirac distribution from quantum physics!) depending on intrinsic agent parameters.
- We added ablations on **prompt length** and on **interaction rounds**, finding results consistent with the original conclusions.
- We added **human data** indicating that while LLMs can perform strongly in persuasive communication, they still **lag behind humans** in simple strategic reasoning required for agentic interaction.

We additionally included semantic analyses and improved visualizations (new figures) to clarify and strengthen the empirical findings.

---

### Meta-Review · Area_Chair_HvKA · 2026-01-05

**Summary:**

The paper introduces Mini-Mafia, a highly simplified four-player social deduction benchmark with fixed night actions and a single discussion/voting phase, designed to isolate LLM capabilities in deception (mafioso), detection (villager), and disclosure (detective). Win rates across model triplets are aggregated into role-specific scores via Bayesian inference. Results highlight counterintuitive size trends and emergent phenomena (name bias, last-speaker advantage).

Reviewers appreciated the controlled setup for isolating social capabilities, the thorough evaluation across models, and interesting emergent findings. However, major concerns centered on the extreme simplification reducing social complexity and realism (limited interaction depth, fixed actions, no multi-round dynamics), insufficient differentiation from prior deception benchmarks (Avalon, Werewolf), lack of human baselines or absolute grounding, limited interpretability of win rates as capability measures, and absence of robustness studies (prompt/round variations).

The authors' rebuttal introduced a theoretical model predicting win probabilities from intrinsic parameters, robustness ablations (prompt length, rounds), human performance data, and deeper strategic analyses. These additions strengthened interpretability and addressed some methodological gaps but did not fully resolve debates on the benchmark's depth, necessity relative to existing work, or real-world relevance.

**Reviewer Concerns:**

Clarifications on robustness and human data were valuable. However, concerns persist on oversimplification limiting insight into complex social reasoning, unclear added value over richer deception benchmarks, and whether measured capabilities generalize beyond the toy setting.

**Reviewer Scores:**

Initial scores were low-to-moderate (three at 2–4, one at 2). Rebuttal yielded positive engagement but no confirmed upward revisions, with consensus remaining skeptical due to scope and novelty questions.

---

### Decision · Program_Chairs · 2026-01-26

Reject